# The modified 30-second chair stand test (m-30s-CST) is more sensitive than handgrip strength in detecting muscle strength changes and predicting physical performance in hospitalized geriatric patients

Walther M. W. H. Sipers[1]*, Martijn J.A. Rothbauer[1,¤a], Isis Ensink[1,¤b], Shannon Röhlinger[2], Audrey H.H. Merry[3]

1 Department of Geriatric Medicine, Zuyderland Medical Center, Sittard-Geleen, the Netherlands,
2 Department of Physical Therapy, Zuyderland Medical Center, Sittard-Geleen, the Netherlands,
3 Department of Epidemiology, Zuyderland Medical Center, Sittard-Geleen, the Netherlands

¤a Current Address: Department of Geriatric Medicine, Meander Medical Center, Amersfoort, the Netherlands;
¤b Current Address: Training Institute for Elderly Care Medicine, Utrecht, the Netherlands
* w.sipers@zuyderland.nl

## Abstract

### Objectives

To compare the modified-30s-Chair-Stand-Test (m-30s-CST) with handgrip strength (HGS) in measuring muscle strength in acutely ill geriatric patients. The aim of this study was to compare the responsiveness and predictive value of the m-30s-CST and HGS for physical performance and two-year mortality in hospitalized geriatric patients.

### Methods

Responsiveness of the m-30s-CST and HGS was assessed in 92 patients (mean age 84±6 years, 53.5% female) by comparing the performance at hospital admission and the day before discharge. These changes were then compared with changes in the ADL-Barthel-Index (ADL-BI) and Short Physical Performance Battery (SPPB).

### Results

The number of repetitions on the m-30s-CST increased significantly during hospitalization in patients who improved on ADL-BI (n=43) and SPPB (n=33) and did not change in those who remained stable or worsened (ADL-BI: n=32 and SPPB: n=26). There was no significant change in HGS in either patients who improved on respectively ADL-BI (n=43) and SPPB (n=41), nor in those who remained stable or worsened (ADL-BI: n=31 and SPPB: n=34). The m-30s-CST showed a trend towards prediction of two-year mortality (AUC 0.609; P=0,071) while HGS did not (AUC 0.573; P=0.221).

**Data availability statement:** All relevant data are within the paper and its Supporting Information file.

**Funding:** The author(s) received no specific funding for this work.

**Competing interests:** The authors have declared that no competing interests exist.

Two-year mortality was significantly higher in geriatric patients (n = 92) with less than 6 repetitions compared with patients with more than 5 repetitions on the m-30s-CST (HR 2.739; CI-95%: 1.173–6.396; P = 0.020). HGS according the EWGSOP-2 criteria was not associated with 2-year mortality (HR 0.969; CI-95%: 0.495–1.900; P = 0.928).

## Conclusion

The m-30s-CST is superior to HGS for assessing changes in muscle strength and serves as a better proxy for physical performance, and is probably a predictor of two-year mortality in hospitalized geriatric patients.

---

## Introduction

Skeletal muscle mass and strength are strong prognostic factors for the functional decline, morbidity, and mortality of acutely ill hospitalized geriatric patients. The European Working Group Sarcopenia Older Peoples-2 (EWGSOP-2) recommends measuring handgrip strength (HGS) or performing the Five times repeated Chair Stand Test (5t-CST) as proxy for muscle strength in diagnosing sarcopenia [1]. In earlier work on this topic, Ensink and coworkers found poor feasibility for the 5t-CST, with only 25% of hospitalized geriatric patients able to perform the test. These findings suggest that the 5t-CST is not a feasible tool for evaluating clinical course and treatment during hospitalization. In contrast, they demonstrated good feasibility and excellent test-retest reliability of the Modified Thirty Seconds Chair Stand Test (m-30s-CST) in acutely ill hospitalized geriatric patients. In addition, significant associations were observed between the m-30s-CST and both the Barthel Index for Activity of Daily Living (ADL-BI) and the Short Physical Performance Battery (SPPB), supporting its value as a proxy for functional status in this population [2]. Additionally, they reported a significant association was found between the m-30s-CST and the ADL-BI and the SPPB in a cross- sectional observation. There appeared to be a weak association between HGS and respectively m-30s-CST, ADL-BI and SPPB [2]. In geriatric medicine, a muscle strength assessment tool must detect changes during hospitalization to provide insight into physical function and outcome. A systematic review and meta-analysis from Aarden et al found that HGS decreased during hospital admission (median admission duration 14.7 days) in electively admitted older patients, but not in acutely admitted patients [3]. Within that second group (n = 343, mean 79 ± 6.6 y, 49% female) they found no significant change in HGS (−0.5 kg; P = 0.08), but a significant improvement in 5t-CST (+0.7; P < 0.001) between admission and 3 months post admission. To the authors' knowledge, no research has been done on the change in muscle strength and its relation to changes in physical performance and self-reliance during hospitalization in acutely ill geriatric patients. Furthermore, information is lacking on the prognostic value of HGS and m-30s-CST in these physically compromised, frail patients.

In this study, we want to investigate how muscle strength, measured with the m-30s-CST and HGS, changes during hospitalization in patients who improve in

ADL-BI and SPPB compared to patients in whom no improvement is found. Additionally, we aim to explore the association between m-30s-CST, HGS and 2-year mortality.

## Materials and methods

### Study sample

All geriatric patients admitted to the acute geriatric ward of the Zuyderland general hospital (the Netherlands) were asked to participate in the study. Recruitment took place over two periods of three months in March 2021 and November 2022. A total of 92 patients were included in the study after admission to the acute care geriatric hospital ward. Inclusion criteria were patients aged above 70 years, a Groningen Frailty Indicator (GFI) score of 4 or greater indicating frailty and having independent mobility (with or without walking aid) before hospital admission [4]. All participants or representatives signed an informed consent form before the start of the study and received written information about the study. Exclusion criteria were being terminally ill with very limited life expectancy, an inability to sit in a standard chair with armrest, not being instructible for performing the tests for any reason or no consent given by patient or representative.

This study complied with the guidelines set out in the Declaration of Helsinki and was approved by the Ethics Committee of Zuyderland and Zuyd Hogeschool, the Netherlands (METCZ20210028).

### Patients' characteristics

Patient characteristics were retrieved from the medical and nursing files. These included sex, age, diagnosis at hospital admission, medical history, body mass index, nutritional status and frailty score. Carlsson Comorbidity Index (CCI) is a validated tool to predict 10-year survival in patients with multiple comorbidities and is used to categorize the comorbidities [5]. The CCI was assessed by a resident of the geriatric department based on the information in the electronic medical file. Bodyweight was measured on a sitting weight scale (SECA, Model 959). Malnutrition was measured using the Short Nutritional Assessment Questionnaire (SNAQ), which is a validated screening instrument for malnutrition. Scores range from 0 to 5; a score of 3 or higher indicates that the patient is malnourished [6]. The frailty score was assessed according to the Groningen Frailty Indicator (GFI) criteria, which covers multiple dimensions, including physical, functional, psychological and social aspects and ranges from 0 to 15: a score of 4 or higher indicates frailty [7]

To study the responsiveness of m-30s-CST and HGS, measurement of these instruments were performed within 3 days after hospitalization and the day before hospital discharge. To study changes in self-reliance and physical performance, respectively the ADL-Barthel Index (ADL-BI) and Short Physical Performance Battery (SPPB) were assessed at these time points. Two-year mortality data were obtained by consulting electronic health records to verify whether patients were alive or deceased, and to document the date of death if applicable. The tests were assessed by the involved researcher, ward resident, physiotherapist or the involved medical student. The tests were carried out in the patients allocated hospital room. Detailed descriptions of the measurement procedures for the m-30s-CST, HGS, ADL-BI and SPPB can be found in Ensink et al (2025) [2].

### ADL-BI and SPPB

We assume that most patients had already experienced some functional decline due to acute illness before hospital admission. Patients were categorized into two groups; those who improved on the ADL-BI and SPPB and those who deteriorated or remained stable between hospital admission and day before discharge. Given this pre-hospital decline, remaining stable on ADL-BI and SPPB can be interpreted as deterioration.

### Two-year mortality after hospitalization

Information on each patient's status was retrieved from the hospital electronic medical file to determine whether patients were still alive or were deceased at two years after the initial hospital admission, including the exact date of death for

survival curve analyses. One researcher retrieved all the information at one time point two years after the inclusion of the last patient.

## Statistics

Data analyses were done using IBM SPSS Statistics version 29. This study was originally powered to test the test-retest reliability [2]. Therefore, we performed a post-hoc analysis to assess the responsiveness of the m-30s-CST compared to changes in the ADL-BI. Based on the available sample sizes (n = 43 improved on the ADL-Barthel Index; n = 33 stable or worsened) and an alpha of 5%, the post-hoc power to detect the observed difference of 3.2 points in the change in m-30s-CST (means ± SD: 3.4 ± 3.3 vs. 0.2 ± 4.1) was 95.6%. Descriptive statistics were used for patient characteristics with means and corresponding standard deviations for continuous variables and percentages for categorical variables. Normal distribution of data (m-30s-CST and HGS) was observed with histograms including skewness and kurtosis and tested using Kolmogorov-Smirnov (KS) test of normality. Repeated Measures ANOVA tests were performed to study changes in m-30s-CST and HGS during hospital stay and in relation with patients' changes on ADL-BI and SPPB (improvement or stable/worsened).

To test the impact of the m-30s-CST and HGS on the 2-year mortality the Kaplan- Meier Curve was performed. Furthermore a Receiver Operating Characteristics curve (ROC) analysis to identify the optimal cut-off value and quantify the test's discriminative power (AUC) of the m-30s-CST and HGS for 2-year mortality. Subsequently Cox proportional hazard ratio (HR) analysis was performed to determine which of these variables could best predict 2-year mortality. Because of the limited number of patients included and the limited number of 'events' (i.e., number of deaths throughout the follow up period), a maximum of three covariates were tested at the same time.

## Results

### Study sample

A total of 92 patients were included in the study. Patient recruitment and exclusion with reasons are shown in the flow chart (S1 Fig).

### Patients' characteristics

The 92 patients that participated in this study, had a mean age of 84 y ± 6SD and a mean GFI score of 6.1 ± 2.9 SD. A total of 49 (53.3%) patients were female.

The baseline patient characteristics, including age, BMI, GFI score, CCI score, SNAQ score, ADL-BI, SPPB, HGS and m-30s-CST were analyzed for differences between patients who improved (n = 43) on the ADL-BI compared to those who remained stable or worsened (n = 33) (Table 1). The results indicated that only the CCI score was significantly higher (P = 0.009) in patients who remained stable or worsened on the ADL-BI compared with the patients who improved during hospital stay (Table 1).

### Responsiveness of the m-30s-CST in relation to ADL-BI

Forty-three patients improved on the ADL-BI during admission. In these patients the number of repetitions on the m-30s-CST increased significantly (P < 0.001) during hospitalization from 3.7 ± 3.3 to 6.1 ± 3.2 repetitions. Thirty-three patients worsened or remained stable on the ADL-BI. In these patients the number of repetitions on the m-30s-CST did not change significantly from 4.4 ± 3.9 to 4.2 ± 4.1 during hospital stay. Changes in the m-30-s-CST differed significantly (P < 0.001) between patients who improved on the ADL-BI and those who worsened or remained stable (Fig 1A and Table 2).

**Table 1. Baseline patient characteristics (n = 76) in the acutely ill hospitalized geriatric patients who improved on the ADL-Barthel Index compared with the patients who remained stable or worsened.**

| Variable | ADL- Barthel Index improved | n | ADL- Barthel Index worsened or stable | n | P-value |
|---|---|---|---|---|---|
| **Age, frailty, and comorbidity at admission** | | n | | n | |
| Female (%) | 56 | 22 | 44 | 17 | 0.580 |
| Male (%) | 57 | 21 | 43 | 16 | |
| Age (years ± SD) | 83 ± 6 | 43 | 84 ± 7 | 33 | 0.133 |
| BMI (mean +SD, kg/m$^2$) | 24.9 ± 3,9 | 43 | 24.5 ± 3.1 | 33 | 0.304 |
| GFI (mean +SD) | 6,3 ± 3,0 | 43 | 5.7 ± 2.9 | 33 | 0.197 |
| CCI (mean +SD) | 5.7 ± 1.8 | 43 | 6.7 ± 2.1 | 33 | 0.009* |
| SNAQ (mean +SD) | 1.6 ± 1.7 | 43 | 1.9 ± 1.7 | 33 | 0.429 |
| **Physical performance** | | | | | |
| SPPB (mean +SD) | 2.8 ± 2.4 | 43 | 2.8 ± 2.5 | 33 | 0.486 |
| ADL-BI (mean +SD) | 11.3 ± 4.3 | 43 | 13.7 ± 4, | 33 | 0.016* |
| **Muscle strength** | | | | | |
| HGS (mean +SD, kg) | 17.7 ± 7.8 | 43 | 18.0 ± 11.2 | 33 | 0.285 |
| m-30s-CST (mean + SD) | 3.7 ± 3.3 | 43 | 4.5 ± 3.8 | 33 | 0.174 |

SD = standard deviation; BMI = Body Mass Index; GFI = Groningen Frailty Index; CCI = Charlson Comorbidity Index; SNAQ = Short Nutritional Assessment Questionnaire; SPPB = Shorth Physical Performance Battery; ADL-BI = Activity of Daily Living Barthel score; HGS = handgrip strength; 5t-CST = 5 times repeated chair stand test, scored according to the SPPB categories; m-30s-CST = modified 30 seconds chair stand test.

* p < 0.05 indicates statistically significant difference.

### Responsiveness of the m-30s-CST in relation to SPPB

### Responsiveness of the HGS in relation to ADL-BI

Forty-three patients improved on the ADL-BI, showing a non-significant change in HGS during hospitalization from 16.7 ± 7.8 to 18.4 ± 8.0 kg. Thirty-three patients worsened or remained stable on the ADL-BI. In these patients the HGS did not change significantly from 18.3 ± 11.5 to 20.0 ± 12.8 during hospital stay. Between the patients who improved on the ADL-BI compared with the patients who worsened or remained stable, the changes in the HGS were not significantly different (P = 0.068) (Fig 1B and Table 2).

### m-30s-CST in relation to SPPB

Thirty-three patients improved on the SPPB, showing a significant increase in m-30s-CST repetitions during hospitalization from 3.2 ± 2.9 to 5.8 ± 2.5 (P < 0.001). Twenty-six patients worsened or remained stable on the SPPB. In these patients the number of repetitions on the m-30s-CST did not change significantly from 3.8 ± 3.4 to 4.0 ± 3.7 during hospital stay. The m-30s-CST difference was highly significant in those who improved on the SPPB versus the patients who worsened or remained stable on the SPPB (P < 0.001) (Fig 2A and Table 2).

### Responsiveness of the HGS in relation to SPPB

Forty-one patients improved on the SPPB. In these patients the HGS increased non-significantly (P = 0.148) during hospitalization from 18.6 ± 10.5 to 20.4 ± 11.9. Thirty-four patients worsened or remained stable on the SPPB, with unchanged HGS during hospitalization (15.5 ± 8.2 versus 15.3 ± 6.8). The HGS difference was not significant in those who improved on the SPPB compared with the patients who worsened or remained stable on the SPPB (P = 0.069) (Fig 2B and Table 2).

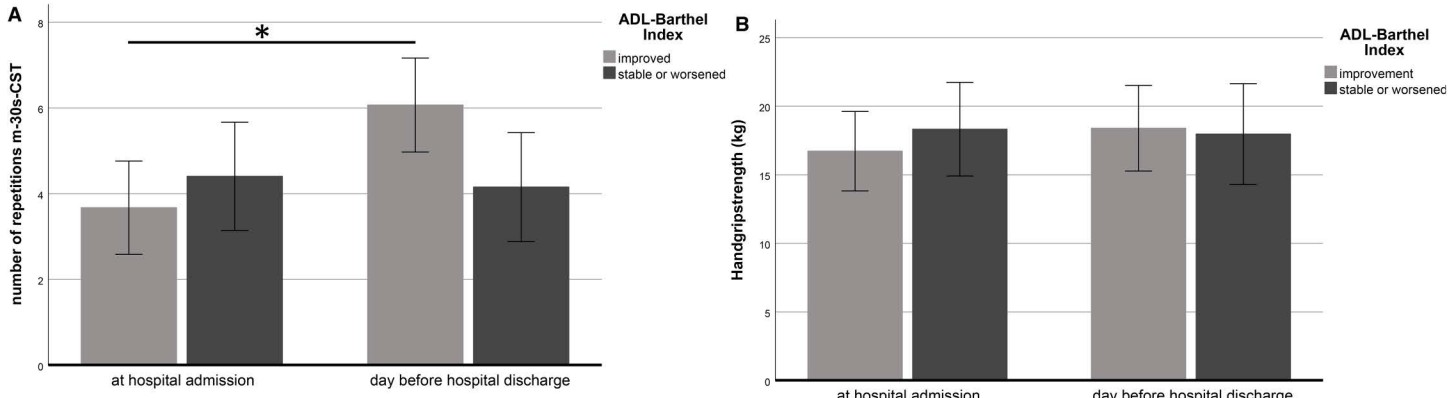

**Fig 1. A. m-30s-CST responsiveness with ADL-BI change.** Changes in number of repetitions on the m-30s-CST in 76 hospitalized geriatric patients who improve compared with the patients who remain stable or worsened on the ADL-BI during hospital stay. * p<0.05 indicates statistically significant difference. **B. HGS responsiveness with ADL-BI change.** Changes in handgrip strength in 76 hospitalized geriatric patients who improve compared with the patients who remain stable or worsened on the ADL-BI during hospital stay.

**Table 2. Changes in in number of repetitions on the m-30s-CST (upper part) and changes in handgrip strength (lower part) in hospitalized geriatric who improve compared with the patients who remain stable or worsened on the ADL-BI and SPPB during hospital stay.**

| m-30s-CST | | n | T<72h after hospital admission | T day before discharge | Mean differences in time m-30s-CST | Mean differences between groups in change in m-30s-CST and |
|---|---|---|---|---|---|---|
| ADL-Barthel Index | improved | 43 | 3.67±3.34 | 6.07±3.23 | P<0.001* | P<0.001* |
| | stable or worse | 32 | 4.41±3.88 | 4.16±4.06 | | |
| SPPB | improved | 33 | 3.18±2.91 | 5.82±2.47 | P<0.001* | P<0.001* |
| | stable or worse | 26 | 3.77±3.42 | 4.04±3.70 | | |
| HGS | | n | T<72h after hospital admission | T day before discharge | Mean differences in time HGS | Mean differences between groups in change in HGS |
| ADL-Barthel Index | improved | 43 | 16.72±7.83 | 18.40±8.03 | P=0.233 | P=0.068 |
| | stable or worse | 31 | 18.32±11.50 | 19.97±12.76 | | |
| SPPB | improved | 41 | 18.61±10.49 | 20.39±11.93 | P=0.148 | P=0.069 |
| | stable or worse | 34 | 15.50±8.18 | 15.29±6.80 | | |

* p<0.05 indicates statistically significant difference.

## Change in performance during hospital stay

**ADL-BI and relation with other characteristics.** Forty-three patients improved and thirty-three patients remained stable or worsened on the ADL-BI. There was no significant relation between age, GFI, SNAQ score between the patients who remained stable or worsened compared to those who improved on the ADL-BI. The patients who remained stable or worsened on the ADL-BI had a significant higher CCI score (6.7±2.1 versus 5.7±1.7; P=0.011) and length of hospital stay (13.8±15.1 versus 6.7±2.1; P=0.003) compared to those who improved on the ADL-BI (S1 Table).

## SPPB and relation with other characteristics

Forty-one patients improved and thirty-five patients remained stable or worsened on the SPPB. There was no significant relation between age, CCI, SNAQ score and length of hospital stay between the patients who remained stable or

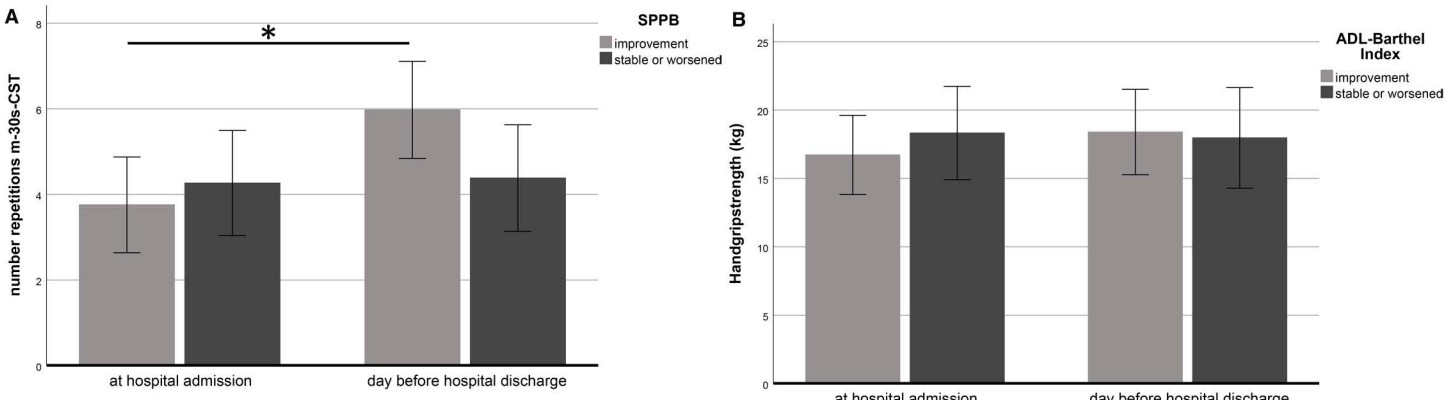

**Fig 2. A. m-30s-CST responsiveness with SPPB change.** Changes in number of repetitions on the m-30s-CST in 59 hospitalized geriatric patients who improve compared with the patients who remain stable or worsened on the SPPB during hospital stay. * p < 0.05 indicates statistically significant difference. **B. HGS responsiveness with SPPB change.** Changes in handgrip strength in 75 hospitalized geriatric patients who improve compared with the patients who remain stable or worsened on the SPPB during hospital stay.

worsened compared to those who improved on the SPPB. The patients who remained stable or worsened on the SPPB had a significant lower GFI score (5.2 ± 2.9 versus 6.7 ± 2.9; P = 0.017) compared to those who improved on the SPPB (S2 Table).

## Muscle strength parameters and two-year mortality

Two years after hospital admission, forty-nine patients of the 92 patients were deceased. The deceased patients had a significantly lower m-30s-CST at hospital admission (3.5 ± 3.0 versus 5.2 ± 4.4, P = 0.014) compared with those who survived (S3 Table). Handgrip strength was not significantly different in patients that died within two years compared with those patients who survived (S3 Table).

The Kaplan Meyer survival curves showed significantly higher mortality rates for the patients who were only able to perform 5 or less compared with the patients who were able to perform 6 or more repetitions on the m-30-CST in hospitalized geriatric patients (Fig 3 and Table 3). These findings were similar for the patients who were able to perform 3 or less compared to 4 or more, 4 or less compared to 5 or more repetitions on the m-30s-CST. There was no significant difference in 2 year survival in patients with low performance on the 5t-CST and handgrip strength compared with patients with a normal performance (Table 3).

Cox proportional hazard ratio analysis was performed on data for n = 92 geriatric patients. Patients with more repetitions on the m-30s-CST (HR 0.898; CI-95%:0.796–1.014; P = 0.082) showed a trend of a lower mortality probability throughout the two year follow-up after hospital admission. Higher HGS (HR 1.007; CI-95%: 0.962–1.054; P = 0.761) was not associated with mortality over the two-year post-admission follow-up. HGS categorized as low (n = 66; mean 13.0 + 6.0 kg) versus normal (n = 26; mean 25.3 + 9.5 kg) according the EWGSOP-2 (female <16 kg and men < 27 kg) did not predict mortality (HR 0.969; CI-95%: 0.495–1.900; P = 0.928). Patients with less than 6 (n = 59; mean 2.0 + 1.8) compared with 6 or more (n = 33; mean 8.6 + 2.5) repetitions on the m-30s-CST had a significant higher mortality risk (HR 2.739; CI-95%: 1.173–6.396; P = 0.020)

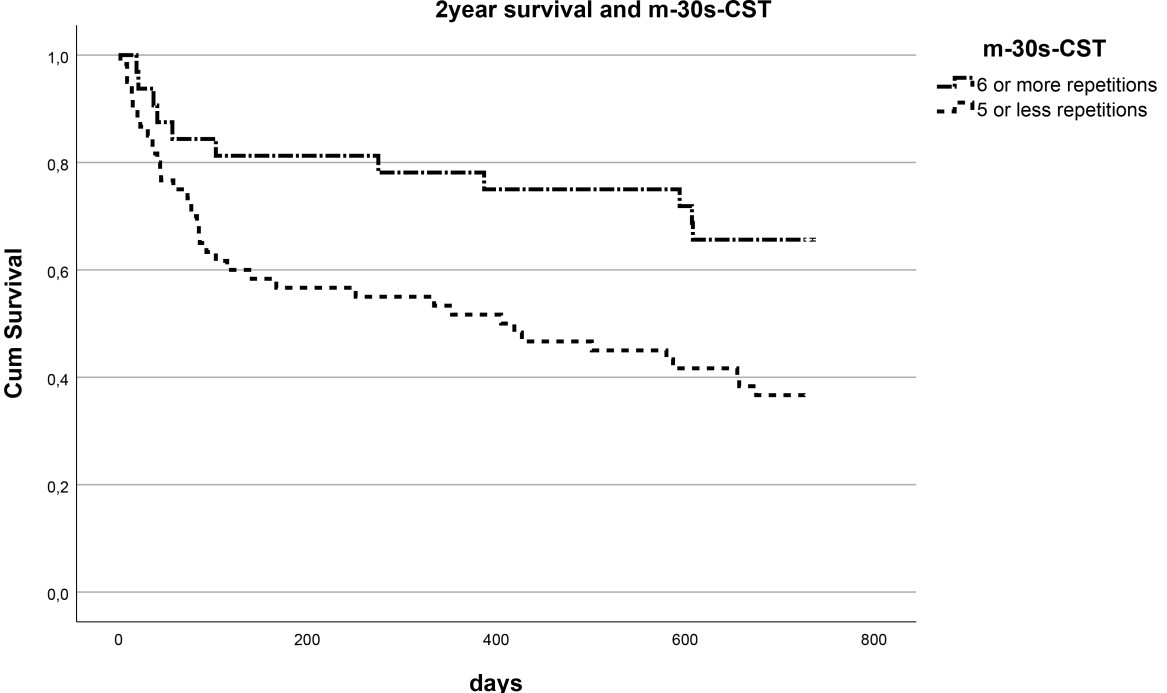

**Fig 3. Survival curve 5 or less versus 6 or more m-30s-CST repetitions.** Kaplan–Meier survival curves comparing 2-year mortality in acutely hospi-talized geriatric patients with low (≤5) versus higher (≥6) m-30s-CST performance (P = 0.009).

**Table 3. Muscle strength cutoffs and 2-year survival.**

| 2-year mortality predictor | Cutoff | n | alive | deceased | Log Rank | Breslow | Tarone-Ware |
|---|---|---|---|---|---|---|---|
| 5t-CST | Normal < 15sec | 8 | 5 | 3 | 0.464 | 0.570 | 0.517 |
| | Poor > 15 sec | 84 | 38 | 46 | | | |
| m-30s-CST | >1 | 67 | 30 | 37 | 0.522 | 0.541 | 0.548 |
| | ≤1 | 25 | 13 | 12 | | | |
| m-30s-CST | >2 | 57 | 27 | 30 | 0.822 | 0.842 | 0.823 |
| | ≤2 | 35 | 16 | 19 | | | |
| m-30-CST | >3 | 45 | 27 | 18 | 0.020* | 0.046* | 0.030* |
| | ≤3 | 47 | 16 | 31 | | | |
| m-30-CST | >4 | 40 | 24 | 16 | 0.022* | 0.023* | 0.022* |
| | ≤4 | 52 | 19 | 33 | | | |
| m-30-CST | >5 | 33 | 22 | 11 | 0.009* | 0.011* | 0.009* |
| | ≤5 | 59 | 21 | 38 | | | |
| HGS (BMI) | normal | 11 | 5 | 6 | 0.950 | 0.990 | 0.969 |
| | low | 81 | 38 | 43 | | | |
| HGS (EWGSOP-2) | Normal (>16 kg/ >27 kg) | 26 | 13 | 13 | 0.690 | 0.683 | 0.686 |

Kaplan–Meier survival curves assessing the association between muscle strength (m-30s-CST and handgrip strength, using multiple cutoff values) and 2-year survival in acutely hospitalized geriatric patients (n = 92).

* p < 0.05 indicates statistically significant difference.

The ROC curve shows that the m-30s-CST has an area under the curve (AUC) of 0.609 (CI 95%: 0.488–0.731) with P = 0.071, showing a trend towards predicting two-year mortality. The ROC curve shows that the HGS has an AUC = 0.573 (CI 95%: 0.455–0.691) with P = 0.221, showing no predictive value of two-year mortality.

## Discussion

In this study, we demonstrate that changes in the m-30s-CST – but not in HGS – during hospital stay are significantly associated with changes in physical performance (SPPB) and self-reliance (ADL-BI) in 92 acutely ill hospitalized geriatric patients. Furthermore, we demonstrated that only the m-30s-CST and not HGS was possibly a weak predictor of two-year mortality. HGS with the cut-off value according to the EWGSOP-2 has no predictive value for 2-year mortality, however the m-30s-CST with a cut-off value of <6 repetitions is a significant predictor of 2 year mortality.

To date, no studies have directly compared the responsiveness of the m-30s-CST and HGS during hospitalization, its associations with changes in SPPB and ADL-BI and predictive value for two-year mortality. In a cross sectional study of older rehabilitation patients (n = 33), a strong correlation was found between the m-30s-CST and a modified ADL-BI, although the responsiveness of the m-30s-CST was not assessed [5]. Another cross-sectional study reported only a weak correlation between HGS and functional performance (SPPB), while changes during hospitalization were not examined [8]. Another study in older surgical patients found a decline in HGS during hospital stay in patients who declined on the KATZ-ADL, this finding differs from our study, likely reflecting differences in patient characteristics and the application of a scale with a different cut-off value [9]. Our finding aligned with prior observations indicating that HGS remained unchanged, whereas knee-extension strength declined in acutely hospitalized patients. However, they did not investigate the association between knee-extension strength and changes in functional performance (ADL-BI or on the Morton Mobility Index) during hospitalization [10].

Our finding that the 5-times Chair Stand Test (5t-CST) was not feasible is consistent with results from a study in geriatric rehabilitation inpatients, due to the patients' limited physical capacity [11]. Contrary to their findings, our results indicated that HGS did not predict mortality, while the modified 30-Second Chair Stand Test (m-30s-CST) demonstrated predictive value. The lack of predictive value of HGS for mortality may reflect the predominance of values below the EWGSOP2 cut-off, reducing discrimination capacity [1]. The modified 30-Second Chair Stand Test (m-30s-CST), adapted from the 5t-CST to address a floor effect, proved a more suitable predictor of mortality in these frail, physically compromised patients. To our knowledge, no studies have investigated the predictive value of the m-30s-CST in terms of mortality.

The m-30s-CST as a measure of muscle strength in the legs is relevant in acutely ill hospitalized geriatric patients because this test, unlike HGS, shows to be sensitive for changes in physical functioning (SPPB) and self-reliance (ADL-BI). Applebaum and coworkers found that in older adult Veterans residing in a long term care hospital only the m-30s-CST and not TUG was significantly associated with patients who fall and numbers of falls over 1 year [12]. Improving and retaining self-reliance with an independent mobility and overall condition is one of the main goals of acutely ill geriatric hospitalized patients [13]. Prioritizing these patient goals underlines the importance of measurements that provide information about changes in muscle strength especially leg muscle strength and the impact on changes in physical functioning during hospital stay.

This study has several possible limitations that should be acknowledged. Of the 371 screened patients, only 92 were ultimately included in the study. In view of the high number of patients that are excluded, there may be selection bias. Because the main reasons for exclusion were logistical and ethical, we believe this does not substantially limit the applicability of the m-30s-CST in the acute hospital setting for geriatric patients who were independently mobile prior to hospitalization. In a small subset of geriatric patients who become (temporarily) wheelchair-bound due to acute illness, HGS may perform better. However, in the majority of patients, our results indicate that change in m-30s-CST performance is a better predictor of in-hospital physical recovery (ADL-BI and SPPB). We do not, however, have specific data on the responsiveness of HGS to changes in physical function during hospitalization among geriatric patients who are (temporarily) wheelchair-bound.

Another possible limitation of the study is that changes in clinical condition during hospitalization, e.g., recovering from a delirium, can have a different impact on handgrip strength and the m-30s-CST, due to different cognitive complexity [14]. Furthermore changes in catabolic state due to treatment of the acute illness, e.g., with severe inflammation, malnutrition

and immobility can impact changes in muscle strength [4]. However, the extent to which catabolism differentially impacts changes in upper-limb (handgrip strength) versus lower-limb (m-30s-CST) muscle strength, and their respective relationships to physical function changes (SPPB and ADL-BI), remains unknown.

A third potential limitation is the uncertainty regarding the premorbid ADL levels, which may reflect chronic impairments, acute illness-related new-onset disability leading to hospitalization, or both, and their impact on changes in HGS and m-30s-CST. Although clinically relevant, this distinction does not affect our primary research question.

Furthermore the predictive power of the m-30s-CST and HGS on 2-year mortality remains uncertain, as the study was not powered for this outcome and we cannot adjust for a number of clinically important confounders, such as the impact of delirium or treatment of the disease. These findings thus require confirmation in larger, adequately powered multicenter studies.

As the focus of this study was on hospitalized acutely ill geriatric patients with prehospital independent mobility, thus concerning frail older patients with high care dependency and low physical performance capacities, caution should be taken when generalizing the findings to other populations. Furthermore a future multicenter study is needed to confirm findings before generalizing the results and including muscle mass in addition to muscle strength as an important characteristic of sarcopenia according to EWGSOP-2 and more functional outcomes like falls, hospital readmissions, recovery of autonomy, resilience, rehabilitation outcome and discharge planning. The strength of this study is its conduct in routine hospital practice with acutely ill geriatric patients. In addition, it is the first study that investigates the responsiveness of HGS and m-30s-CST during hospitalization, their relation with changes in self-reliance (ADL-BI) and physical performance (SPPB).

## Conclusion

The m-30s-CST, unlike HGS, is responsive to changes in physical performance (SPPB) and self-reliance (ADL-BI) during hospitalization in acutely ill hospitalized geriatric patients. Moreover, the m-30s-CST may impact two year mortality, suggesting prognostic potential beyond that of HSG. In daily geriatric ward practice, the m-30s-CST provides a simple bedside tool to monitor lower-extremity strength and functional recovery, informing decisions on rehabilitation intensity and discharge planning. Further research should clarify the predictive value of m-30s-CST measurements during hospital stay on the impact on rehabilitation and discharge planning and other clinical relevant outcomes such as falls, readmissions, autonomy recovery, resilience and mortality.

## Supporting information

**S1 Fig. Flowchart of patient inclusion.**
(PDF)

**S1 Table. Impact of patient characteristics on the patients who improved compared to the patients who remained stable or worsened on the Barthel Index in hospitalized geriatric patients (n = 76).** * p < 0.05 indicates statistically significant difference.
(PDF)

**S2 Table. Impact of patient characteristics on the patients who improved compared to the patients who remained stable or worsened on the SPPB in hospitalized geriatric patients (N = 76).** * p < 0.05 indicates statistically significant difference.
(PDF)

**S3 Table: Muscle strength, physical function, age, comorbidity, frailty, nutrition and BMI versus 2-year survival in acutely hospitalized geriatric patients (n = 92).** * p < 0.05 indicates statistically significant difference.
(PDF)

**S1 Data. Minimal dataset.**
(DOCX)

## Author contributions

**Conceptualization:** Walther M.W.H. Sipers.

**Data curation:** Isis Ensink, Martijn J.A. Rothbauer.

**Formal analysis:** Walther M.W.H. Sipers, Isis Ensink, Audrey H.H. Merry.

**Investigation:** Isis Ensink, Martijn J.A. Rothbauer, Shannon Röhlinger.

**Methodology:** Walther M.W.H. Sipers, Audrey H.H. Merry.

**Project administration:** Isis Ensink, Martijn J.A. Rothbauer, Shannon Röhlinger.

**Resources:** Isis Ensink, Martijn J.A. Rothbauer.

**Supervision:** Walther M.W.H. Sipers.

**Validation:** Walther M.W.H. Sipers, Isis Ensink, Audrey H.H. Merry.

**Writing – original draft:** Walther M.W.H. Sipers.

**Writing – review & editing:** Isis Ensink, Martijn J.A. Rothbauer, Shannon Röhlinger, Audrey H.H. Merry.

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
