## [Decision Letter · Decision Letter 0]

3 Nov 2025

PONE-D-25-42511The modified 30-second chair stand test (m-30s-CST) is more sensitive than handgrip strength in detecting muscle strength changes and predicting physical performance and mortality in hospitalized geriatric patientsPLOS ONE

Dear Dr. Sipers,

Thank you for submitting your manuscript to PLOS ONE. After careful consideration, we feel that it has merit but does not fully meet PLOS ONE’s publication criteria as it currently stands. Therefore, we invite you to submit a revised version of the manuscript that addresses the points raised during the review process.

The limited sample size, potential selection bias, lack of adjustment for key confounders, and the empirical definition of m-30s-CST cut-offs weaken the strength of the conclusions. A major revision is required to improve the statistical robustness, clarify the definition of functional outcomes, and better highlight the clinical implications for geriatric practice.

We look forward to receiving your revised manuscript.

Kind regards,

Francesco Curcio, M.D., Ph.D.

Academic Editor

PLOS ONE

Journal Requirements:

2. We note that your Data Availability Statement is currently as follows:

“All relevant data are within the manuscript and its Supporting Information files.”

6. Please upload separately your supporting tables with the file type 'Supporting Information'. Please ensure that each Supporting Information file has a legend listed in the manuscript after the references list. Please see our Supporting Information guidelines for more information: http://journals.plos.org/plosone/s/supporting-information.

**Additional Editor Comments:**

The study addresses an important and clinically relevant question in geriatric hospital care and is clearly written. However, several methodological and analytical issues need to be addressed before the manuscript can be considered for publication. In particular, the limited sample size, potential selection bias, lack of adjustment for key confounders, and the empirical definition of m-30s-CST cut-offs weaken the strength of the conclusions. A major revision is required to improve the statistical robustness, clarify the definition of functional outcomes, and better highlight the clinical implications for geriatric practice.

Reviewers' comments:

Reviewer's Responses to Questions

**Comments to the Author**

1. Is the manuscript technically sound, and do the data support the conclusions?

Reviewer #1: Yes

Reviewer #2: Yes

2. Has the statistical analysis been performed appropriately and rigorously? 

Reviewer #1: Yes

Reviewer #2: Yes

3. Have the authors made all data underlying the findings in their manuscript fully available?

Reviewer #1: Yes

Reviewer #2: Yes

4. Is the manuscript presented in an intelligible fashion and written in standard English?

Reviewer #1: Yes

Reviewer #2: Yes

5. Review Comments to the Author

Reviewer #1: The authors compare the modified-30s-Chair-Stand-Test (m-30s-CST) with handgrip strength (HGS) in measuring muscle strength in acutely ill geriatric patients. Responsiveness of the m-30s-CST and HGS was assessed in 92 patients (mean age 84±6 y, 53.5% female) by comparing the performance at hospital admission and the day before discharge. These changes were then compared with changes in the ADL-Barthel-Index (ADL-BI) and Short Physical Performance Battery (SPPB). The number of repetitions on the m-30s-CST increased significantly during hospitalization in patients who improved on ADL-BI (n=43) and SPPB (n= 33) and did not change in those who remained stable or worsened (ADL-BI: n= 32 and SPPB: n= 26). There was no significant change in HGS in either patients who improved on respectively ADL-BI (n=43) and SPPB (n=41), nor in those who remained stable or worsened (ADL-BI: n=31 and SPPB: n= 34). Two-year mortality was significantly higher in geriatric patients with low performance on the m-30s-CST. HGS was not associated with 2-year mortality.

The manuscript is interesting but I have some concern about the clinical status of the patients.

1) Frailty: you are considered only physical frailty and not multidimensional frailty. Please see and discuss: Liguori I et al. Validation of "(fr)AGILE": a quick tool to identify multidimensional frailty in the elderly. BMC Geriatr. 2020 Sep 29;20(1):375.

2) Catabolic syndrome: what about parameters characterizing the “catabolic syndrome” in these patients? Please see and discuss: Curcio F et al. The reverse metabolic syndrome in the elderly: Is it a "catabolic" syndrome? Aging Clin Exp Res. 2018 Jun;30(6):547-554.

Reviewer #2: The study addresses a clinically relevant question in hospital geriatrics: which strength test better reflects changes in physical performance and predicts mortality in acutely ill older adults. The modified 30-second chair stand test (m-30s-CST) was compared with handgrip strength (HGS) in 92 patients (mean age 84 years). The authors found that the m-30s-CST was more responsive to in-hospital changes and a stronger predictor of two-year mortality, while HGS showed no significant associations.

This is a well-written and interesting paper. It provides practical insight into functional assessment in frail geriatric patients and highlights the limitations of handgrip testing in this setting.

Some methodological limitations, however, should be considered:

• The sample size is relatively small and derived from a single center, which limits generalizability.

• A large number of exclusions may have introduced selection bias. This also underlines a limitation in the applicability of the m-30s-CST in the acute hospital setting.

• It is not entirely clear which level of ADL function is considered at admission — the one assessed at the time of hospitalization (which may already be unrealistic and influenced by multiple confounding factors) or the premorbid status immediately before the acute event. In any case, the definition of "incident disability" (new-onset disability) related to hospitalization remains conceptually problematic for the study’s interpretation.

• The Cox analysis included only a few covariates and did not adjust for potential confounders such as length of stay, immobilization, or rehabilitation interventions.

• No measure of muscle mass was included, preventing an integrated analysis of structural sarcopenia.

• Mortality cut-offs for the m-30s-CST were defined empirically rather than validated. Given this, I recommend performing a ROC curve analysis to identify the optimal cut-off value and quantify the test’s discriminative power (AUC). This would strengthen the statistical validity of the conclusions and allow direct comparison with handgrip strength.

• Functional outcomes beyond mortality (e.g., falls, readmissions, recovery of autonomy) were not assessed, although these would be clinically meaningful in this population.

• Finally, the discussion and conclusions could more clearly highlight the key clinical message and what this study adds to daily geriatric practice.

6. PLOS authors have the option to publish the peer review history of their article (what does this mean? ). If published, this will include your full peer review and any attached files.

**Do you want your identity to be public for this peer review?** For information about this choice, including consent withdrawal, please see our Privacy Policy .

Reviewer #1: No

Reviewer #2: No

---

## [Author Response · Author response to Decision Letter 1]

26 Jan 2026

The above information and question is intended for the editor

---

## [Decision Letter · Decision Letter 1]

2 Feb 2026

The modified 30-second chair stand test (m-30s-CST) is more sensitive than handgrip strength in detecting muscle strength changes and predicting physical performance in hospitalized geriatric patients

PONE-D-25-42511R1

Dear Dr. Walther Sipers ,

We’re pleased to inform you that your manuscript has been judged scientifically suitable for publication and will be formally accepted for publication once it meets all outstanding technical requirements.

Kind regards,

Francesco Curcio, M.D., Ph.D.

Academic Editor

PLOS One

Additional Editor Comments (optional):

Reviewers' comments:

Reviewer's Responses to Questions

**Comments to the Author**

1. If the authors have adequately addressed your comments raised in a previous round of review and you feel that this manuscript is now acceptable for publication, you may indicate that here to bypass the “Comments to the Author” section, enter your conflict of interest statement in the “Confidential to Editor” section, and submit your "Accept" recommendation.

Reviewer #1: All comments have been addressed

Reviewer #2: All comments have been addressed

2. Is the manuscript technically sound, and do the data support the conclusions?

Reviewer #1: Yes

Reviewer #2: Yes

3. Has the statistical analysis been performed appropriately and rigorously? 

Reviewer #1: Yes

Reviewer #2: Yes

4. Have the authors made all data underlying the findings in their manuscript fully available?

Reviewer #1: Yes

Reviewer #2: Yes

5. Is the manuscript presented in an intelligible fashion and written in standard English?

Reviewer #1: Yes

Reviewer #2: Yes

6. Review Comments to the Author

Reviewer #1: The manuscript is really improved. All questions have been arised. The manuscript merits to be published.

Reviewer #2: The manuscript is improved and points of criticism were analysed and discussed in the limitation of the study section. I found the manuscript suitable for publication.

7. PLOS authors have the option to publish the peer review history of their article (what does this mean? ). If published, this will include your full peer review and any attached files.

**Do you want your identity to be public for this peer review?** For information about this choice, including consent withdrawal, please see our Privacy Policy .

Reviewer #1: No

Reviewer #2: No

---

## [Editor Report · Acceptance letter]

PONE-D-25-42511R1

PLOS One

Dear Dr. Sipers,

I'm pleased to inform you that your manuscript has been deemed suitable for publication in PLOS One. Congratulations! Your manuscript is now being handed over to our production team.

Kind regards,

on behalf of

Dr. Francesco Curcio

Academic Editor

PLOS One